# MoME: Mixture of Matryoshka Experts for Audio-Visual Speech Recognition

**Umberto Cappellazzo**
Imperial College London

**Minsu Kim**
Meta AI

**Pingchuan Ma**
Meta AI

**Honglie Chen**
Meta AI

**Xubo Liu**
Meta AI

**Stavros Petridis**
Imperial College London
NatWest AI Research

**Maja Pantic**
Imperial College London
NatWest AI Research

## Abstract

Large language models (LLMs) have recently shown strong potential in audio-visual speech recognition (AVSR), but their high computational demands and sensitivity to token granularity limit their practicality in resource-constrained settings. Token compression methods can reduce inference cost, but they require fixing a compression rate in advance and produce a single fixed-length output, offering no flexibility to balance information density and efficiency at inference time. Matryoshka representation learning (MRL) addresses this by enabling a single model to operate across multiple token granularities, allowing compression rates to be adjusted dynamically. However, current MRL-based methods treat each scale independently during training, limiting cross-scale generalization, robustness at high compression, and interpretability. To overcome these limitations, we propose MoME (Mixture of Matryoshka Experts), a novel framework that integrates sparse Mixture-of-Experts (MoE) into MRL-based LLMs for AVSR. MoME augments a frozen LLM with top-k routed and shared experts, allowing dynamic capacity allocation across scales and modalities. A shared router promotes consistent expert activation across granularities, enabling compressed sequences to benefit from representations learned at lower compression. Experiments on LRS2 and LRS3 demonstrate that MoME achieves state-of-the-art performance across AVSR, ASR, and VSR tasks, while requiring significantly fewer parameters and maintaining robustness under noise. MoME unifies the adaptability of MRL with the efficiency of MoE, offering a scalable and interpretable solution for resource-aware speech recognition.

## 1 Introduction

To overcome the inherent limitations of Auditory Speech Recognition (ASR) [1, 2, 3], which typically suffers from degraded performance in acoustically challenging environments (e.g., crowded areas or subways), **Audio-Visual Speech Recognition** (AVSR) [4, 5, 6] has attracted substantial attention in the speech processing community. By incorporating an additional visual modality, such as lip movements, that remains unaffected by acoustic noise, AVSR seeks to enhance the robustness and accuracy of speech recognition systems under noisy conditions.

Recent advancements in Multimodal Large Language Models (MLLMs) have shown that integrating modalities such as vision and speech significantly broadens the scope and capability of LLMs, achieving state-of-the-art performance across a range of tasks [7, 8, 9, 10, 11, 12, 13, 14, 15, 16]. Following this trend, recent literature has explored the incorporation of LLMs into ASR and AVSR tasks and achieved promising performance [17, 18, 19, 20, 21, 22]. However, a critical challenge

39th Conference on Neural Information Processing Systems (NeurIPS 2025).

with these models is their **token hunger**: they tend to perform better when given fine-grained, dense token representations of the input. This is particularly problematic in the audio-visual speech domain, where inputs are long in duration and have a higher temporal resolution compared to text, resulting in a much larger number of input tokens than in text-only settings [23]. To address the computational burden, it is common practice to *reduce the number of tokens* before feeding them to the LLM. Several strategies have been proposed for this purpose, including token concatenation along the hidden dimension [14, 24, 25, 26], resampling-based modules like query transformers [27, 28, 29, 30, 31], average pooling [23, 32, 33], CTC-based compression [34], and compression via speech-text alignment [35]. While effective, these methods suffer from a *key limitation*: they require to specify a **fixed token compression rate** in advance, thus producing a **single fixed-length token sequence**, which may not be optimal for all inputs or deployment scenarios and prevents dynamic control over the trade-off between information fidelity and computational efficiency.

To mitigate these challenges, **Matryoshka**-based LLMs have emerged as a flexible and efficient framework for *elastic inference*, initially applied to vision-language tasks [32, 36], and more recently extended to AVSR [23]. Inspired by the **Matryoshka Representation Learning** (MRL) principle [37], these approaches train models across multiple token granuralities, allowing the number of audio and visual input tokens to be dynamically adjusted at inference time, depending on resource availability or task-specific constraints. However, current Matryoshka models rely on uniform, monolithic representations at each scale and treat each resolution independently during training. This lack of inter-scale interaction forces the model to compromise between generality and specialization, often yielding suboptimal performance at higher rates, where less input information is preserved. Moreover, a detailed analysis of the interaction between audio and visual tokens across

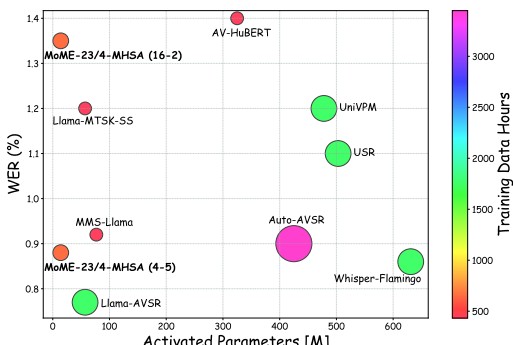

Figure 1: Comparison of MoME with SOTA methods in terms of WER, number of activated parameters, and training data hours on LRS3 dataset. MoME achieves performance parity with or outperforms recent AVSR models while training on a lesser amount of hours, activating fewer parameters and catering to user's resource constraints with a single set of model weights.

different scales remains largely unexplored, limiting the interpretability and adaptability of these models.

To address these limitations, we propose **Mixture of Matryoshka Experts** (MoME), a novel module that integrates sparse *Mixture-of-Experts* (MoE) into the Matryoshka framework for AVSR. MoME introduces *top-k routed experts* into each layer of a frozen pretrained LLM, allowing the model to dynamically allocate computational capacity across token granularities and modalities. Each MoME module includes a *shared router* that processes both audio and visual tokens and selects specialized experts per token. Moreover, inspired by recent models like DeepSeekMoE [38] and Llama 4 [39], we also incorporate one or two *shared experts* to capture global, cross-modal, and scale-invariant knowledge that is useful across all scales. A key feature of MoME is that, during inference, the router consistently activates similar expert subsets across scales. This implicit alignment enables sequences at higher compression rates to leverage expert pathways shaped by richer, lower-compression sequences, promoting knowledge transfer and improving representation quality at all scales. When combined with the shared experts, this design enhances both specialization and generalization, leading to stronger performance across a wide range of inference-time budgets. *To the best of our knowledge, this is the first work to propose a unified framework that encompasses both MoE and MRL paradigms*.

MoME consistently outperforms prior Matryoshka-based methods as well as independently trained models across AVSR, ASR, and Visual Speech Recognition (VSR) tasks on LRS2 [40] and LRS3 [41] benchmarks. Thanks to its sparsely activated architecture, MoME requires significantly fewer parameters during inference than competing baselines. In addition, MoME offers architectural flexibility, as it can be seamlessly inserted at multiple points within the LLM, as well as strong robustness in noisy scenarios, outperforming alternative methods. Finally, we conduct extensive ablations on the key components of MoME, such as the number of routed and shared experts. We

further provide visualization analyses that shed light on how audio and visual tokens interact across scales, offering new insights into cross-modal and cross-scale alignment. Altogether, MoME unifies the scale-aware adaptability of MRL with the efficiency and modularity of MoE, resulting in a single model capable of resource-aware inference with strong interpretability.

## 2 Related Work

### 2.1 Mixture of Experts

Mixture-of-Experts (MoE) models have recently demonstrated significant advancements [42, 38, 43, 44, 45, 46, 47]. The key idea is to replace a standard Feed-Forward Network (FFN) layer with an MoE layer, which comprises multiple FFN experts and a learnable (usually sparse) routing mechanism. By selectively activating only a subset of experts for each input, MoE architectures enhance performance while preserving computational efficiency. Recent works have focused on making MoE models more efficient and stable via co-upclying [48, 49, 7, 50], sharing parameters across layers [51, 52], and adjusting the number of experts and those to activate dynamically [53]. While most of the works focus on MoEs as a pretraining strategy, they can also be used in parameter-efficient finetuning by keeping the Transformer backbone frozen and only update a small number of parameters [13, 54, 55, 56]. Finally, in the audio-visual domain, recent works have demonstrated that MoEs can efficiently scale model capacity and optimally process audio and visual data [57, 58, 59, 21].

### 2.2 Matryoshka Representation Learning

The challenge of developing flexible representations adaptable to a spectrum of downstream tasks with varying computational demands has been addressed by Matryoshka Representation Learning (MRL) [37, 60, 61, 62, 63, 64]. This method encodes information at different granularities within a unified, high-dimensional feature vector produced by a neural network. Recently, the MRL paradigm has been used in vision-language and audio-visual LLMs to learn a hierarchy of representation granularities at the token sequence level [36, 32, 23], enabling adaptive deployment per computational constraints.

### 2.3 Audio-Visual Speech Recognition

Early deep learning approaches to AVSR focused on designing modality-specific encoders and fusion strategies for integrating audio and visual inputs [65, 66, 67, 68]. The introduction of Transformers [69] led to substantial performance gains [70, 71, 72], which spurred further research into multimodal modeling, including self-supervised learning [73, 74, 75, 76], knowledge distillation from ASR to AVSR [77, 78], and leveraging cross-modal complementarity [79, 80, 81].

With the rise of LLMs, recent works [20, 28, 19, 15, 22, 16, 21] have explored integrating audio and audio-visual speech recognition capabilities into pre-trained LLMs. In LLM-based AVSR, a key challenge is the computational cost of processing long speech and visual sequences. Llama-AVSR [14] addressed this by compressing audio-visual features at fixed rates, demonstrating reduced compute demands with limited performance degradation. More recently, Llama-MTSK [23] uses MRL [36, 32] to enable elastic inference, allowing a single model to process multiple token granularities at test time. Concurrently, MMS-Llama [29] proposed an adaptive compression strategy based on speaking rate using a Q-Former [82]. However, both Llama-AVSR and MMS-Llama require retraining for each specific token rate to match the desired GPU memory usages, limiting deployment flexibility. This hinders the practicality of using LLM-based AVSR models, which usually require a GPU with a large memory capacity. Inspired by the granularity-adaptive processing capabilities of Llama-MTSK, we propose MoME, which combines MRL [37] with sparse MoE layers inserted into a frozen LLM. Unlike prior methods, MoME facilitates knowledge transfer by reusing expert activations across token granularities and incorporating shared experts that encode scale-invariant information. This design significantly mitigates performance drops at high compression rates, offering a single, resource-adaptive model that maintains strong performance across a wide range of compute budgets.

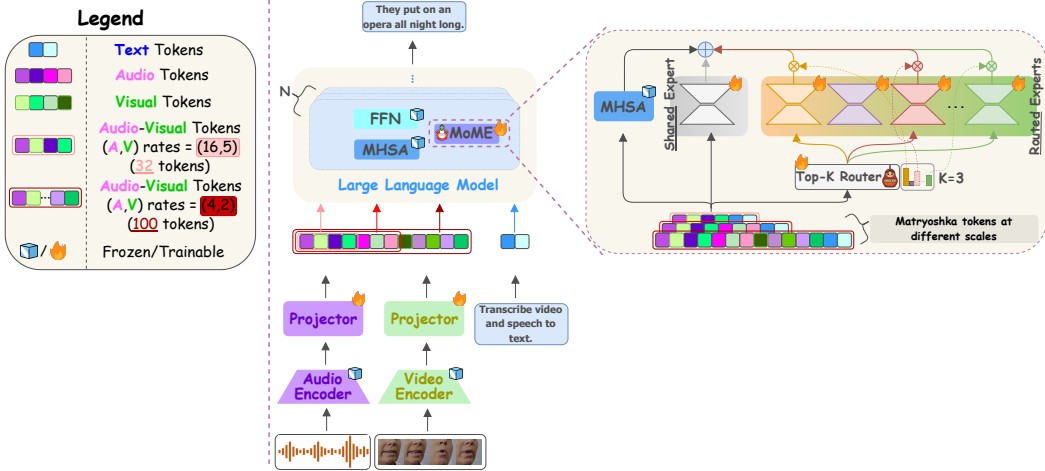

Figure 2: Overview of our proposed MoME module. We start by producing audio-visual tokens at different scales via modality-specific pre-trained encoders and projectors. Each Matryoshka sequence goes through MoME, which can be placed parallel to multiple modules within each LLM layer (parallel to the MHSA module in the Figure). Each MoME module comprises a *top-k router*, which sparsely activates a subset of *routed experts*, and a pool of *shared experts* to capture scale-invariant knowledge.

## 3 Methodology

### 3.1 Audio-Visual Matryoshka Token Sequences

Given an audio waveform $\mathbf{a}$ and its corresponding lip movements video $\mathbf{v}$, we process them using a pre-trained audio encoder (*e.g.*, Whisper [83]) and a pre-trained video encoder (*e.g.*, AV-HuBERT [73]) to yield audio and visual token sequences, $\mathbf{Z}^a$ and $\mathbf{Z}^v$, respectively.

When processing audio-visual tokens with an LLM, reducing token granularity is essential for lowering computational cost and improving inference efficiency. This is particularly important in AVSR, where both modalities exhibit temporal continuity, resulting in redundant information. However, most token compression methods require a fixed compression rate to be set in advance, limiting the ability to dynamically balance recognition performance and resource constraints. While finer-grained tokens improve recognition accuracy, they substantially increase inference cost due to the quadratic complexity of Transformers, reducing their practicality in constrained settings.

To address this, we adopt the MRL framework [32], which allows audio-visual granularity to be flexibly controlled at inference time based on specific requirements. We generate token sequences at varying granularities by applying different compression rates to the audio and visual streams, enabling the model to learn to operate across a spectrum of coarse-to-fine resolutions within a single architecture. This approach has proven successful in both vision-language and audio-visual MLLMs [36, 23, 32].

Concretely, following [23], in training we apply $\mathsf{G}$ audio compression rates $\{a_1, a_2, \cdots, a_\mathsf{G}\}$ to obtain audio token sequences at multiple scales $[\mathbf{Z}^{a_1}, \mathbf{Z}^{a_2}, \cdots, \mathbf{Z}^{a_\mathsf{G}}]$, and $\mathsf{L}$ video compression rates $\{v_1, v_2, \cdots, v_\mathsf{L}\}$ to obtain $[\mathbf{Z}^{v_1}, \mathbf{Z}^{v_2}, \cdots, \mathbf{Z}^{v_\mathsf{L}}]$. Each sequence is passed through modality-specific projection layers to match the LLM's embedding dimension. We then generate $\mathsf{G} \cdot \mathsf{L}$ audio-visual token sequences by concatenating all audio-video rate combinations along the temporal axis. Finally, textual tokens (i.e., instructions and ground truth) are appended to each sequence. We denote a sequence with audio rate $a_i$ and video rate $v_j$ as $\mathbf{Z}^{ij}$.

### 3.2 Mixture of Matryoshka Experts (MoME) Module

Even though MRL enables elastic inference by supporting multiple token granularities, some performance degradation at lower token scales is inevitable due to the information loss introduced by compression [23]. To mitigate this, we leverage the fact that all coarse-to-fine token sequences are accessible during MRL training. Our proposed solution, **Mixture of Matryoshka Experts** (MoME),

introduces a set of *routed* and *shared* experts trained jointly across token granularities. Each expert is responsible for modeling specific phonetic patterns and is exposed to both high- and low-resolution representations of the same utterance during training. This design allows experts to learn fine-grained (high-resolution) features that can later be reused when processing compressed (low-resolution) tokens at inference time. By aligning expert training across scales, MoME promotes cross-granularity knowledge transfer and improves the robustness of the model under aggressive compression.

The MoME module adapts the standard Mixture-of-Experts (MoE) layer to the Matryoshka framework through the following key design modifications. **(1)** In typical MoE-based LLMs, each FFN layer is replaced with an MoE layer and trained over large-scale data [84, 44, 85, 86, 43, 42, 46, 47]. In contrast, we insert MoME modules in parallel with the existing LLM layers, allowing efficient fine-tuning of a frozen pretrained backbone. To minimize the number of active parameters during inference, we design the experts with a small bottleneck dimension, pushing it down to $1$ in the extreme case. This aligns with the *shallow brain hypothesis* [87], which posits that cognition involves not only deep sequential processing (analogous to LLMs) but also many parallel shallow modules (as realized by MoME experts). Furthermore, reducing the bottleneck dimension allows us to activate more experts per token, akin to the *fine-grained expert segmentation* proposed in [38]. **(2)** We further include one or more shared experts, which are always active regardless of input, to capture global and cross-scale knowledge. This guarantees that even highly compressed inputs benefit from stable, general representations, following insights from prior work [38]. **(3)** Crucially, both the experts and router in each MoME module are shared across all Matryoshka sequences. This design encourages the router to activate similar subsets of routed experts across different granularities, creating implicit alignment. As a result, lower-resolution sequences can leverage the representational capacity learned from higher-resolution inputs, promoting robustness and knowledge transfer across scales.

Each MoME module consists of $N_r$ *routed* experts, which are sparsely activated based on the *router*'s scores, and $N_s$ *shared* experts, which process deterministically each input token. Each expert follows a bottleneck structure similar to adapter and LoRA experts [88, 54, 89, 90, 13, 91], where the first linear layer serves as downsampler, followed by a non-linear activation (e.g., GELU), and uprojected back to the LLM hidden size. Then, the output of the MoME module can be expressed as:

$$\text{MoME}(\mathbf{H}_l^{ij}) = \sum_{n=1}^{N_s} E_n\left(\mathbf{H}_l^{ij}\right) + \sum_{n=N_s+1}^{N_s+N_r} \left(\text{g}_n E_n(\mathbf{H}_l^{ij})\right), \tag{1}$$

$$\text{g}_n = \begin{cases} \text{s}_n, & \text{s}_n \in \text{Top-k}(\{\text{s}_p | N_s + 1 \le p \le N_s + N_r\}, K), \\ 0, & \text{otherwise}, \end{cases} \tag{2}$$

$$\text{s}_n = \text{Softmax}_n(\mathbf{H}_l^{ij^T} \mathbf{W}_l^n), \tag{3}$$

where $\mathbf{H}_l^{ij}$ is the input to the MoME block corresponding to the $ij$-th Matryoshka sequence and $l$-th LLM layer, and it can be the output of the Multi-Head Self-Attention (MHSA) module if it is placed parallel to the FFN module, otherwise it is the input of the LLM layer after the layer normalization, $E_n(\cdot)$ is the $n$-th expert, $\text{g}_n$ represents the gate value for $n$-th expert, $\text{s}_n$ denotes the token-to-expert affinity scores, Top-k$(\cdot, K)$ denotes the set comprising the $K$ highest affinity scores among those calculated for the all $N_r$ routed experts, and $\mathbf{W}_l^n$ represents the router's weights (in our case a linear layer) corresponding to the $n$-th expert in the $l$-th layer. Note that $\text{g}_n$ is sparse, indicating that only $K$ out of $N_r$ gate values are non-zero, with $K \ll N_r$. This sparsity property ensures computational efficiency within a MoME module, *i.e.*, each token will be assigned to and computed via only $K$ routed experts.

The proposed MoME module can be placed in parallel to three locations within a Transformer layer of the LLM: **1)** the MHSA module, **2)** the FFN, and **3)** the whole Transformer layer (see section B in the Appendix for the visualization of the three variants). In the following, we consider inserting MoME in parallel with the FFN module, and the extension to the other two cases can be straightforwardly achieved. Thus, with the MoME insertion, the $l$-th Transformer layer of the LLM for the sequence $\mathbf{Z}^{ij}$ is:

$$\mathbf{H}_l^{ij} = \text{MHSA}(\mathbf{Z}_{l-1}^{ij}) + \mathbf{Z}_{l-1}^{ij}, \tag{4}$$

$$\mathbf{Z}_l^{ij} = \text{FFN}(\mathbf{H}_l^{ij}) + \text{MoME}(\mathbf{H}_l^{ij}) + \mathbf{H}_l^{ij}, \tag{5}$$

where $\mathbf{H}_l^{ij}$ is the hidden state after the $l$-th attention module , and $\mathbf{Z}_l^{ij}$ is the output hidden state after the $l$-th Transformer layer. For brevity, we omit the layer normalization in the above formulations.

Our model is trained by averaging the auto-regressive next token prediction loss for each audio-visual scale $ij$ for each input data. The LLM predicts the response $\mathbf{Y} = \{y_s\}_{s=1}^{S}$ conditioned on the multimodal input tokens, where $S$ represents the number of tokens of the ground truth transcription to be generated. Accordingly, for each Matryoshka audio-visual representation $\mathbf{Z}^{ij}$, the probability of the target $\mathbf{Y}$ is computed by $p(\mathbf{Y}|\mathbf{Z}^{ij}) = \prod_{s=1}^{S} p_\theta(y_s|\mathbf{Z}^{ij}, y_{<s})$, where $y_{<s}$ is the generated output sequence up to token $s-1$, and $\theta$ is the trainable parameters. The final objective is the average over all the audio-visual token scales: $\mathcal{L}_{LM} = -\frac{1}{\text{G}\cdot\text{L}} \sum_{i=1}^{\text{G}} \sum_{j=1}^{\text{L}} \log p(\mathbf{Y}|\mathbf{Z}^{ij}) \cdot c_{ij}$, where $c_{ij}$ is a parameter that controls the importance of each audio-visual sequence, and is set to 1 following previous works [32, 36, 23]. We experiment with different weights in Section C.1 of the Appendix. In addition to this, to avoid the risk of router collapse, where the majority of tokens are assigned to only a few experts, we employ a load balancing loss $\mathcal{L}_B$ [45] for each audio-visual sequence $\mathbf{Z}^{ij}$. To compute it, we multiply the fraction of tokens $f_n$ routed to one expert $E_n$ with the total routing probability $P_n$ allocated to $E_n$ for one batch and sum it across the number of routed experts $N_r$ as follows: $\mathcal{L}_B = N_r \cdot \sum_{n=N_s+1}^{N_s+N_r} f_n P_n$. This loss is scaled by a factor $0.01$ following [45, 42].

## 4 Experiments

### 4.1 Experiment Settings

**Datasets**. We train and evaluate MoME on LRS2 [40] and LRS3 [41] datasets. LRS2 includes 225 hours of video clips. LRS3 contains 433 hours of transcribed English video clips.

**Tasks**. We study the AVSR task for our main experiments on LRS2 and LRS3, and we also report the results for the ASR and VSR tasks on LRS3.

**Model Details**. We use AV-HuBERT Large [73] as the visual encoder and Whisper (small and medium versions) [83] as the audio encoder. The projectors consist of two linear layers with ReLU activation in between. For the LLM backbone, we adopt three variants from the Llama 3 family [92]: Llama 3.2-1B, Llama 3.2-3B, and Llama 3.1-8B. Each MoME module comprises a router, implemented as a linear layer, which activates a sparse subset of routed experts per input token using top-k selection, along with one or two shared experts that are always active. We also investigate different insertion points for MoME within the LLM architecture, resulting in three variants: 1) parallel insertion to the FFN module, 2) parallel insertion to the MHSA module, and 3) parallel insertion to the entire LLM layer. For example, MoME-23/4-MHSA denotes a configuration where MoME is inserted parallel to the MHSA module, using 23 routed experts with 4 active per token (i.e., top-4). For the main experiments, we primarily use a single routed expert. Further training and preprocessing details are provided in the Appendix, Section 4.1.

**Audio-Visual Granularities**. For a fair comparison with previous methods, we apply the same compression rates as in [23]. The compression rates are chosen in a coarse-to-fine manner, in order to encompass a range of tokens sequences that trade-off between efficiency and performance at inference. For ASR, the compression rates are {4, 8, 12, 16, 20}. For VSR, we apply {1, 2, 3, 4, 5}. For AVSR, we apply audio rates in {4, 16} and video rates in {2, 5}, leading to 4 audio-visual configurations. For the main experiments, we compress the tokens via average pooling.

**Baselines**. As shown in Table 2, we compare our proposed MoME with multiple methods. **Fixed-rate methods**: 1) Llama-AVSR [14] is the first AVSR LLM-based model, achieving sota results on LRS3. This approach is trained on each audio-visual scale separately, leading to as many models as the number of scales. 2) Similar to this, we also include the baseline MoME-23/4-MHSA-I, which is trained independently on each fixed scale. Both approaches *do not provide* elastic inference within a single model. **Matryoshka methods**: we include a recent work, Llama-MTSK [23], which is the adaptation of Llama-AVSR to the Matryoshka setting. Llama-MTSK supports three configurations: 1) *MS* uses a single Multi-Scale LoRA module for all scales, 2) *SS* uses a single Scale-Specific LoRA module for each scale, and 3) *MSS* combines *SS* and *MS* together. More baselines are used in the bubble plot 1 (see Section 4.2).

### 4.2 Main Results

**AVSR Results (1)**. Table 1 summarizes the AVSR performance of our proposed MoME variants FFN, MHSA, and LAYER. For a fair comparison with prior work, we use Whisper-small as the audio

Table 1: Main results for AVSR on LRS2 and LRS3 in terms of WER (%) across different (audio-visual) compression rates (e.g., (4,2)). [‡] Llama-AVSR [14] and MoME-23/4-MHSA-I involves training separate models for each pair of audio-visual rates. The active parameter count does not include the projector parameters since they are the same for all the reported methods (around 8.1M).

| Method | Active Params | LRS2 Dataset | | | | Active Params | LRS3 Dataset | | | |
|---|---|---|---|---|---|---|---|---|---|---|
| | | (4,2) | (4,5) | (16,2) | (16,5) | | (4,2) | (4,5) | (16,2) | (16,5) |
| Llama-AVSR [14][‡] | 27.5M | 4.1 | 4.5 | 5.3 | 8.1 | 6.8M | 2.4 | 2.8 | 3.3 | 4.1 |
| Llama-MTSK MS [23] | 27.5M | 4.8 | 5.9 | 6.4 | 8.9 | 8.1M | 2.6 | 2.7 | 3.7 | 4.1 |
| Llama-MTSK SS [23] | 27.5M | 3.4 | 4.7 | 4.8 | 6.4 | 8.1M | 2.3 | 2.2 | 3.3 | 3.6 |
| Llama-MTSK MSS [23] | 55.0M | 3.6 | 4.8 | 6.1 | 9.0 | 13.6M | 2.4 | 2.4 | 3.2 | 3.5 |
| MoME-23/4-MHSA-I[‡] | 12.7M | 3.2 | 3.1 | 4.9 | 5.3 | 3.5M | 2.1 | 1.9 | 3.3 | 3.7 |
| **MoME-23/4**-FFN | 12.7M | 3.2 | 3.1 | 4.5 | 4.6 | 3.5M | 2.1 | 2.2 | 4.0 | 4.0 |
| **MoME-23/4**-MHSA | 12.7M | 2.9 | 3.0 | **4.2** | 4.3 | 3.5M | 1.8 | **1.7** | **2.9** | **2.9** |
| **MoME-23/4**-LAYER | 12.7M | **2.7** | **2.7** | **4.2** | **4.2** | 3.5M | **1.5** | 1.8 | 3.1 | 3.2 |
| **MoME-23/4**-LAYER | **2.3M** | 3.0 | 3.2 | 4.3 | 4.7 | **0.9M** | 2.0 | 2.2 | 3.2 | 3.7 |

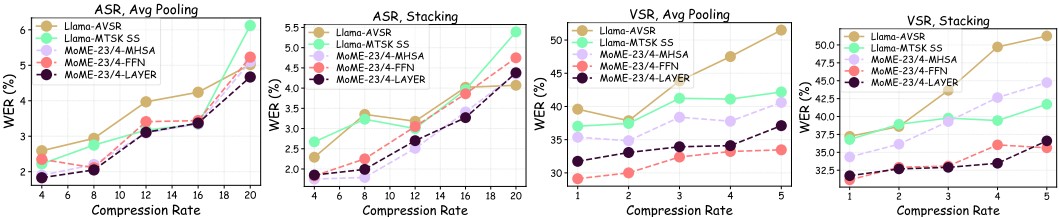

Figure 3: MoME-23/4 results for VSR and ASR tasks on the LRS3 dataset.

encoder, and Llama 3.2-1B and Llama 3.2-3B as the LLMs for LRS3 and LRS2, respectively. **(1)** In the fixed-rate setting, MoME-23/4-MHSA-I outperforms Llama-AVSR by a significant margin across all compression rates, highlighting the superiority of MoME over LoRA as a parameter-efficient adaptation method. **(2)** All three MoME variants consistently outperform Llama-MTSK, with notably lower WER degradation at harsher compression rates. Among the three configurations, inserting MoME parallel to the entire LLM layer or to the MHSA module yields the best performance, particularly the former at higher granularities (e.g., token rates (4,2) and (4,5)). Furthermore, MoME activates consistently fewer parameters thanks to the sparse activation of the experts. **(3)** MoME also outperforms models trained independently at each audio-visual scale, demonstrating not only its support for elastic inference within a single model but also its strong competitive accuracy. **(4)** To further minimize inference cost, we reduce the expert bottleneck dimension to 1, resulting in only 2.3M and 0.9M active MoME parameters for LRS2 and LRS3, respectively. Even under this extremely parameter-efficient setting, MoME-23/4-LAYER exhibits minimal performance degradation, confirming MoME's robustness and scalability.

**AVSR Results (2).** Figure 1 presents a comparison between MoME and recent state-of-the-art methods on the LRS3 benchmark using a bubble chart. Baselines include UniVPM [93], USR [94], Whisper-Flamingo [77], Llama-AVSR [14], Llama-MTSK [23], Auto-AVSR [78], AV-HuBERT [73], and MMS-Llama [29]. Following [14, 23], we use Llama 3.1-8B as LLM and Whisper-medium. The results show that MoME (evaluated at A-V token rates of (4,5) and (16,2)) achieves competitive WER results while activating significantly fewer parameters and requiring fewer training data hours, all under one suite of weights. More extensive results can be found in Section C of the Appendix.

Table 2: AVSR WER results on LRS2 under different acoustic noise levels.

| Method | SNR (dB) | | | | |
|---|---|---|---|---|---|
| | 7.5 | 5 | 2.5 | 0 | -5 |
| Llama-AVSR [14] | 5.6 | 7.1 | 10.6 | 11.8 | 41.8 |
| Llama-MTSK MS [23] | 6.2 | 8.0 | 13.0 | 12.4 | 44.9 |
| **MoME-23/4-LAYER** | 4.8 | 6.4 | 9.6 | 9.6 | 32.6 |

**ASR and VSR Results.** To validate the effectiveness of MoME in unimodal settings, we evaluate its performance on the ASR and VSR tasks. Following [23], we apply average pooling and "stacking" compression, where the latter compresses input by concatenating consecutive tokens along the hidden dimension. As shown in Figure 3, our MoME methods remain effective also when dealing with

Table 3: Ablation on the optimal number of shared and routed experts of MoME-MHSA on LRS2.

| Routed Experts | Shared Expert | Expert Size | Top-k | Compression Rates (A,V) | | | |
|---|---|---|---|---|---|---|---|
| | | | | (4,2) | (4,5) | (16,2) | (16,5) |
| 1 | 0 | 48 | / | 3.4 | 3.4 | 4.9 | 5.1 |
| 4 | 0 | 24 | 2 | 3.3 | 3.3 | 4.8 | 5.0 |
| 4 | 1 | 24 | 2 | 3.2 | 3.2 | 4.4 | 4.7 |
| 7 | 1 | 24 | 2 | 3.2 | 3.3 | 4.5 | 4.7 |
| 15 | 1 | 24 | 3 | 2.9 | 3.0 | 4.3 | 4.5 |
| 23 | 1 | 12 | 4 | 2.9 | 3.0 | 4.2 | 4.3 |
| 23 | 2 | 12 | 4 | 2.8 | 3.0 | 4.1 | 4.7 |
| 23 | 3 | 12 | 4 | 2.9 | 3.0 | 4.2 | 4.6 |

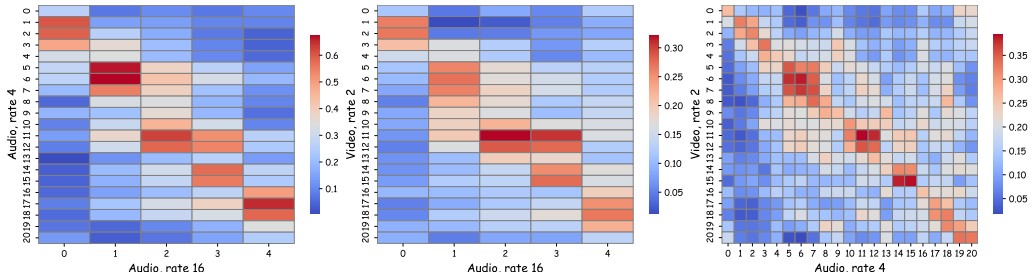

Figure 4: Intra-modality and cross-modality correlation matrices for MoME-15/3 trained on LRS2. We study the sentence: "*it's a long way from home*".

tokens coming from a single modality. In particular, we observe consistent gains across all scales for the FFN and LAYER variants for the VSR task.

## 4.3 Ablation Studies

**Noisy Setting**. To evaluate the robustness of MoME in noisy conditions, we perform inference with varying levels of babble noise from the NOISEX dataset [95]. As shown in Table 2, our MoME-23/4-LAYER configuration exhibits greater resilience to noise compared to prior methods, with particularly stronger gains in highly degraded scenarios.

**Optimal Number of *Shared/Routed* Experts**. In Table 3, we analyze the impact of varying the number of routed and shared experts in the MoME-MHSA configuration. Following the approach in [38], increasing the number of top-k activated routed experts requires proportionally reducing the bottleneck size of each expert to maintain the same number of active parameters during inference. We derive the following insights: **(1)** *The use of a shared expert is beneficial*. When comparing the second and third rows in the table (with and without shared experts, respectively), we observe a consistent reduction in WER across all compression rates when a shared expert is included. This supports the hypothesis that shared experts help capture scale-invariant knowledge. For the configuration with 23 routed experts, adding a second shared expert yields slight improvements at some rates but results in degradation at the (16,5) token rate, along with an increase in active parameters. Therefore, we use a single shared expert in all main experiments. **(2)** Regarding routed experts, *performance generally improves as more routed experts are activated*, suggesting that increased expert diversity enhances the model's representational capacity, even when the total parameter budget remains fixed.

**Optimal Number of *Activated* Experts (k)**. To analyze the impact of top-k, we conduct ablation experiments using the MoME-8/k-MHSA configuration with a single shared expert on the LRS2 AVSR task. The number of activated experts (i.e., k) varies while keeping the number of routed experts fixed at 8. Table 4 shows that increasing k yields moderate gains, especially at higher compression rates. However, these improvements come at the expense of increased computation, which contradicts our design objective of **Sparse** MoE. Therefore, we conclude that using a small k, typically 1 or 2, offers a practical trade-off between efficiency and performance. This choice preserves sparsity while retaining most of the accuracy benefits of broader expert activation.

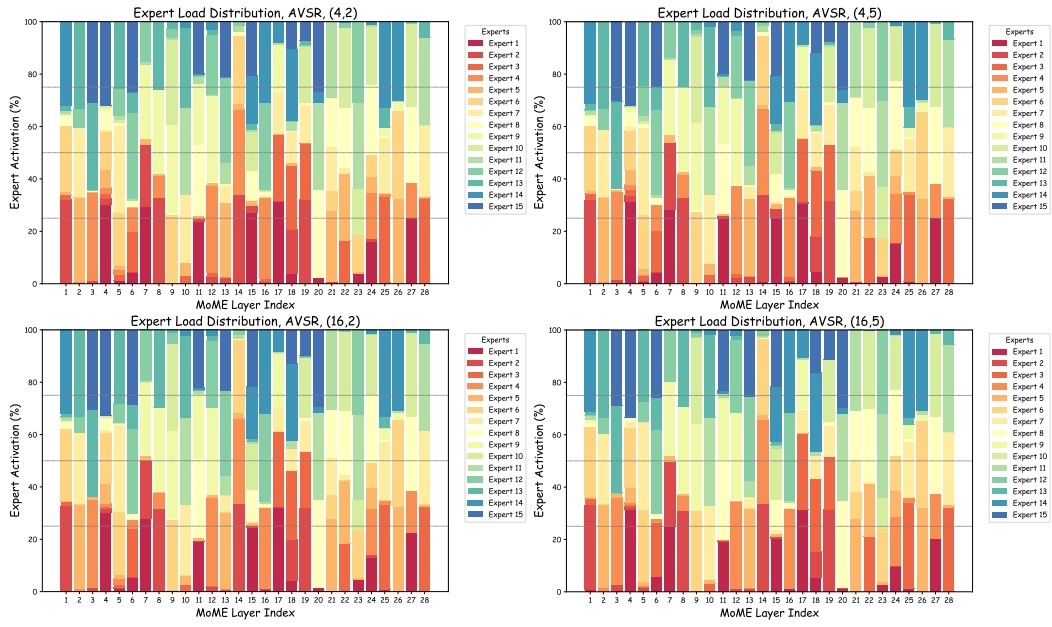

Figure 5: MoME-15/3-MHSA expert activation analysis across multiple scales and layers on LRS2.

## 4.4 Visualization Analysis

We analyze how MoME processes audio-visual speech tokens at different granularities through two complementary visualizations: **(1)** expert activation patterns across scales and layers, and **(2)** similarity matrices between speech representations at varying granularities. Additional results can be found in the Appendix (see Section D).

**Cross-Modal Matryoshka Token Analysis**. We visualize the similarity matrix between token sequences at different granularities. Specifically, we compare video tokens with compression rates of 2 and 5, as well as audio tokens at compression rates of 4 and 16 before being fed to the LLM. We present two types of similarity matrices: 1) **intra-modality** similarity matrices, where we compare tokens of the same modality at different compression rates (e.g., the 21 audio tokens from the sequence with a compression ratio of 4 are compared to the 5 audio tokens with a compression ratio of 16), and 2) **cross-modality** similarity matrices, where we compare tokens from one modality with those from the other. Figure 4 reveals a strong linear correlation between speech features across different granularities, as indicated by high cosine similarity values along the diagonal. On average, each token in sequences with a higher compression ratio correlates mainly with approximately two to three of the tokens coming from less compressed sequences. These findings suggest that when more tokens are available, they tend to encode similar redundant information, which is later condensed into a single token in sequences with higher compression rates. Moreover, they indicate that audio and video tokens of different granularities tend to cluster together rather than being learned independently, a behavior encouraged and reinforced by MoME, which promotes cross-scale alignment through shared expert routing and consistent expert activation across resolutions.

**Expert Activation Analysis**. Figure 5 illustrates the layer-wise expert activation distribution across different scales for the MoME-15/3-MHSA configuration on the LRS2 dataset. Across all settings, we observe a consistent pattern: the same subset of experts tends to be activated across different token granularities, demonstrating strong alignment in routing behavior. This confirms the intended effect of using a shared router and shared experts, which promotes cross-scale consistency in expert utilization. At the same time, the set of activated experts varies significantly from layer to layer, ensuring that all experts are utilized across the network. This layer-wise diversity in routing strategies allows MoME to avoid redundancy and encourages specialization among experts, while still maintaining alignment across input scales. Together, these results validate MoME's ability to balance scale-invariant knowledge sharing with depth-wise expert diversity, which is crucial for achieving robust and efficient performance across a wide range of compression levels.

Table 4: Ablation on the optimal top-k value.

| Top-k | (4,2) | (4,5) | (16,2) | (16,5) |
|-------|-------|-------|--------|--------|
| 1 | 3.3 | 3.3 | 4.6 | 4.7 |
| 2 | 3.2 | 3.3 | 4.5 | 4.7 |
| 4 | 3.2 | 3.2 | 4.5 | 4.6 |
| 6 | 3.2 | 3.2 | 4.5 | 4.5 |
| 8 | 3.1 | 3.2 | 4.4 | 4.5 |

Table 5: Inference time and token per second (TPS) across different compression ratios.

| Ratio | Tokens | Inf. Time (s) | TPS |
|-------|--------|---------------|-----|
| (1,1) | 1673 | 12.75 | 7.76 |
| (4,2) | 560 | 8.04 | 12.90 |
| (16,5) | **184** | **6.74** | **14.17** |

## 4.5 Computation Cost Analysis

In Figure 6, we illustrate the benefits of MoME in terms of TFLOPs and inference costs. Compared to the uncompressed case (i.e., 1673 tokens), MoME enables elastic inference by allowing users to select compression rates based on their computational constraints. By increasing the audio-visual compression rates, we reduce the number of tokens processed by the LLM, and thus the TFLOPs, by up to 8x when applying audio-visual compression rates of (16,5) (184 tokens). Despite this substantial reduction in TFLOPs, the resulting increase in WER remains modest.

In addition to this, we include the actual inference time and generated tokens per second (TPS) measured on an NVIDIA L40 46GB GPU with MoME-15/3-MHSA. As shown in Table 5, for a 23-second speech input, the results demonstrate a significant reduction in inference time as compression ratios increase. For instance, a compression ratio of (16,5) reduced the inference time to 6.74 seconds for transcribing 23 seconds of speech, compared to 12.75 seconds in the no-compression case. This confirms that higher compression rates lead to *faster inference*, *lower GPU memory usage*, and *reduced computational load*, while still *maintaining strong performance* thanks to MoME's expert routing mechanism.

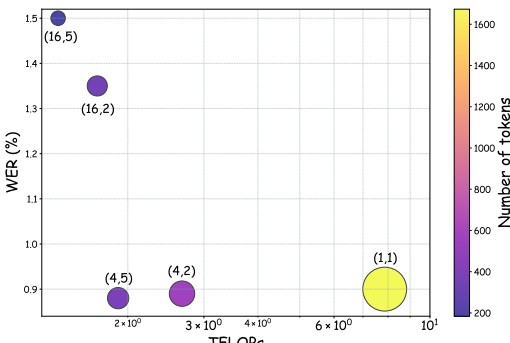

Figure 6: Comparison of MoME-23/4-MHSA in terms of number of audio-visual processed tokens, achieved WER, and TFLOPS.

## 5 Conclusion

We introduce MoME, a framework for audio-visual speech recognition that combines MRL with MoE to enable resource-adaptive inference across token granularities. By using a shared router and shared experts, MoME promotes expert alignment across scales, allowing compressed sequences to benefit from richer representations and maintaining strong performance even at high compression levels. MoME outperforms prior Matryoshka-based and fixed-scale models on AVSR, ASR, and VSR tasks across LRS2 and LRS3, while using significantly fewer parameters and training data hours. It also supports extremely parameter-efficient fine-tuning and demonstrates robustness in noisy conditions. Ablation and visualization analyses further validate MoME's ability to capture cross-scale and cross-modal interactions, making it a scalable and efficient solution for LLM-based AVSR. Although developed for audio-visual speech processing, MoME is expected to be broadly applicable to other multimodal domains, such as vision-language tasks, with minimal adaptation.

## Acknowledgments

All data collection, processing, and experiments were conducted by Imperial College London.

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

Figure 7: Overview of MoME's placement strategies. It supports parallel insertion to: **1) MHSA module**, **2) FFN** module, and **3)** whole **LLM layer**. Layer normalizations are omitted for simplicity. Extensive experiments with the three configurations can be found in the main paper and in the Appendix.

# Appendix

# A    Experiment Details

## A.1    Data Licenses

LRS3 [41] is licensed under CC BY-NC-ND 4.0. LRS2 [40] allows for academic, non-commercial research.

## A.2    Pre-Processing

We follow [78, 78, 23] for the pre-processing of the datasets. For the video modality, we crop the mouth region of interests (ROIs) through a bounding box of $96 \times 96$. Each frame is normalised by subtracting the mean and dividing by the standard deviation of the training set. Audio data only undergo z-normalisation per utterance.

## A.3    Training Details

Following [14, 78], we augment visual inputs through horizontal flipping, random cropping, and adaptive time masking, while for audio we only apply adaptive time masking. For training, similar to [78], we sample bubble noise from the NOISEX dataset [95] using a uniform distribution from the range [-5, 0, 5, 10, 15, 20, $\infty$] dB and add it to the clean speech signal. We define the textual prompts as in [14]: "`Transcribe {task_prompt} to text.`", where `task_prompt` $\in$ {"`speech`", "`video`", "`speech and video`"}. We train our model for $10$ epochs with the AdamW optimizer with cosine annealing scheduler and weight decay set to $0.1$ using NVIDIA H200 GPUs. The learning rate is set to 1e-3 for ASR and AVSR tasks, and 5e-4 for VSR.

## A.4    Inference Details

For decoding, we use beam search with a beam width of $15$ and temperature of $0.6$.

# B    MoME's Insertion Variants

As discussed in Section 3.2, MoME offers high flexibility in that it can be inserted in multiple points within the LLM. Specifically, it supports three variants: **(1)** parallel insertion to the **MHSA** module, **(2)** parallel insertion to the **FFN** module, and **(3)** parallel insertion to the whole **LLM layer**. Figure 7 illustrates these three configurations.

# C    Additional AVSR Experiments

In Figure 1 of the main paper, we compare MoME-23/4-MHSA with several state-of-the-art methods on the LRS3 dataset. To complete our analysis, we report in Table 6 the performance of multiple

Table 6: AVSR results on LRS2 and LRS3 in terms of WER (%) across different (audio-visual) compression rates (e.g., (4,2)). All methods use Whisper medium as audio encoder and Llama 3.1-8B as LLM. ‡ Llama-AVSR [14] involves training separate models for each pair of audio-visual rates.

| Method | Active Params | LRS2 Dataset | | | | LRS3 Dataset | | | |
|---|---|---|---|---|---|---|---|---|---|
| | | $(4,2)$ | $(4,5)$ | $(16,2)$ | $(16,5)$ | $(4,2)$ | $(4,5)$ | $(16,2)$ | $(16,5)$ |
| Llama-AVSR [14]‡ | 27.3M | 2.4 | 2.2 | 2.9 | 3.3 | 0.9 | 0.9 | 1.6 | 2.1 |
| Llama-MTSK MS [23] | 27.3M | 2.1 | 2.3 | 2.9 | 3.2 | 1.0 | 1.1 | 1.5 | 1.6 |
| Llama-MTSK SS [23] | 27.3M | 2.4 | 2.1 | 2.9 | 2.9 | 0.9 | 1.0 | 1.7 | 1.8 |
| Llama-MTSK MSS [23] | 54.5M | 2.4 | 2.5 | 3.2 | 3.4 | 1.2 | 1.0 | 1.5 | 1.6 |
| **MoME-23/4**-MHSA | 14.1M | **1.9** | **1.8** | 2.3 | **2.2** | 0.9 | 0.9 | **1.3** | 1.5 |
| **MoME-23/4**-LAYER | 14.1M | **1.9** | 1.9 | **2.0** | **2.2** | 0.9 | 0.9 | 1.4 | **1.4** |
| **MoME-23/4**-MHSA | **3.5M** | 2.1 | 2.0 | 2.8 | 3.0 | 1.0 | 1.1 | 1.5 | 2.0 |

MoME configurations across all four scales on both LRS2 and LRS3. We observe that the two configurations with $14.1$M parameters consistently outperform prior methods, showing significantly less degradation at lower scales. Furthermore, even when reducing the bottleneck dimension of each expert to $1$, resulting in just $3.5$M active parameters, MoME still achieves strong performance while minimizing parameter usage. These results further highlight MoME's superiority over existing methods and its robustness across scales.

### C.1 Additional Ablations on MoME

In Section 4.3, we ablate key components of MoME, including the number of routed experts, the use of shared experts, and its robustness under noisy conditions. In this section, we focus on the role of **1)** the *shared router* and *2)* the *Matryoshka weights* applied to the token sequences.

MoME uses a router shared across all granularities to activate similar subsets of experts, thereby promoting implicit alignment. To ablate this design choice, we introduce a granularity-specific router, assigning a separate router to each scale while keeping the experts shared. We refer to this variant as **disjoint routers** (DR). As shown in Table 7, using separate routers leads to poorer performance, with greater WER degradation at higher compression rates. This is attributed to the loss of cross-scale alignment enabled by the shared router.

As discussed in Section 3.2, MoME is trained over all the audio-visual token scales:

$$\mathcal{L}_{LM} = -\frac{1}{\mathsf{G} \cdot \mathsf{L}} \sum_{i=1}^{\mathsf{G}} \sum_{j=1}^{\mathsf{L}} \log p(\mathbf{Y}|\mathbf{Z}^{ij}) \cdot c_{ij}.$$

The weight coefficients $c_{ij}$ are set to 1 across all scales, following prior work [32, 36, 23]. To try to improve performance at lower scales, we experiment with **non-uniform weights** (NUW) by increasing the training loss contribution of lower scales, setting $c_{i,j} = [1, 1, 1.5, 2]$. This design emphasizes lower scales during training. However, as shown in Table 7, the NUW variant does not improve performance at lower scales and consistently degrades results at higher scales such as (4, 2) and (4, 5). Consequently, in our main experiments, we adopt uniform weighting by setting $c_{i,j} = 1$ for all scales.

## D  Additional Visualization Analyses

In Section 4.4, we provide several visual analyses to understand how MoME learns tokens at multiple scales. Here, we include additional examples in this direction.

### D.1 Additional Cross-Modal Matryoshka Token Analyses

We include additional correlation matrices in Figure 8. Specifically, we analyze the sentence "*it's a long way from home*" from the LRS2 dataset using the MoME-15/3 configuration. Consistent with

Table 7: Ablation on the use of 1) **disjoint routers** (DR) and 2) **non-uniform weights** (NUW) on the LRS2 and LRS3 datasets for the MoME-15/3-LAYER configuration.

| Method | Compression Rates (A,V) | | | |
|---|---|---|---|---|
| | $(4,2)$ | $(4,5)$ | $(16,2)$ | $(16,5)$ |
| **LRS2 Dataset** | | | | |
| MoME-15/3-LAYER | 2.9 | 2.9 | 4.0 | 4.8 |
| DR | 2.9 | 2.9 | 4.7 | 5.0 |
| NUW | 4.0 | 3.7 | 4.0 | 4.9 |
| **LRS3 Dataset** | | | | |
| MoME-15/3-LAYER | 1.8 | 1.9 | 3.0 | 3.4 |
| DR | 1.9 | 2.0 | 3.2 | 3.7 |
| NUW | 2.4 | 2.2 | 3.5 | 3.6 |

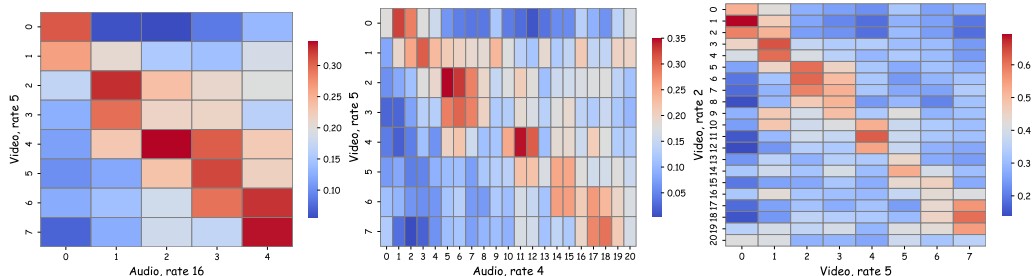

Figure 8: Additional **intra-modality** and **cross-modality** correlation matrices for MoME-15/3 trained on LRS2. We study the sentence: "*it's a long way from home*".

previous results, we observe a strong linear correlation between token sequences across different granularities and modalities (i.e., high cosine similarity values along the diagonal). Thanks to the use of a shared router and shared experts, MoME encourages token sequences at lower granularities to attend to specific portion of sequences at higher granularities. This mechanism helps mitigate performance degradation at lower granularities.

### D.2 Additional Expert Activation Analysis

Figure 9 shows the layer-wise expert activation distribution across different scales for the MoME-23/4-FFN configuration on the LRS2 dataset. Similar to the results in Section 4.4, MoME tends to activate the same experts at a given layer index across the four different scales. Additionally, MoME activates different pools of experts from layer to layer, indicating expert heterogeneity. These results suggest that MoME exhibits consistent behavior across different configurations (e.g., 15 vs. 23 experts, MHSA vs. FFN).

## E  Inference Cost Comparison

We compare MoME with Llama-AVSR [14] and Llama-MTSK [23] in terms of inference cost. Specifically, we compare the three methods in terms of inference time and number of generated tokens per second (TPS). We average over 4 speech inputs with an audio/visual compression rate of (4,2). As reported in Table 8, MoME-15/3-MHSA shows a modest increase in inference time (around 1.25x) and a corresponding reduction in tokens generated per second, primarily due to router overhead and expert dispatching latency, as expected in sparse MoE-based systems, while outperforming by a significant margin the other two methods. However, the primary design goal of MoME is not faster inference, but rather: **1)** Substantially reduced memory requirements via support for high compression rates with strong performance and within the same model, **2)** Robustness under noise, and **3)** Improved interpretability through expert specialization and shared routing. In conclusion, we believe that this moderate inference overhead is justified by the gains in scalability, generalization, and parameter efficiency, particularly for deployment in memory-constrained or

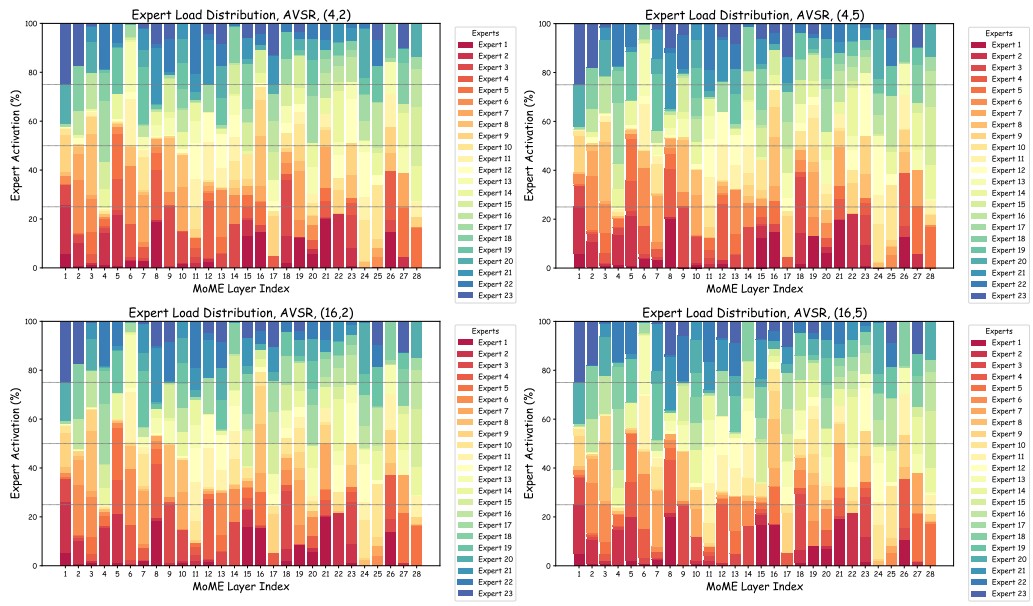

Figure 9: MoME-23/4-FFN expert activation analysis across multiple scales and layers on LRS2.

Table 8: Inference cost comparison between our proposed MoME, Llama-AVSR and Llama-MTSK in terms of inference time and generated tokens per second (TPS).

| Method | Inference Time (s) | TPS | WER ↓ |
|---|---|---|---|
| Llama-AVSR [14] | **2.39** | **15.85** | 4.1 |
| Llama-MTSK [23] | 2.51 | 15.09 | 3.6 |
| MoME-15/3-MHSA | 2.98 | 12.98 | **2.9** |

multi-resolution scenarios. Nonetheless, optimizing router execution remains an interesting avenue for further improving inference efficiency.

## F MoME's Extension to Other Multimodal Tasks

MoME is designed as a parameter-efficient fine-tuning module that operates in parallel with frozen layers of a pretrained LLM or encoder. To apply MoME to other multimodal tasks (e.g., image–text retrieval), two key conditions should be met: **(1)** the presence of a pretrained encoder or LLM backbone that can remain frozen during fine-tuning, and **(2)** input modalities (e.g., image tokens) whose token granularity impacts performance and efficiency, allowing for compression.

Beyond these considerations, MoME can be integrated with minimal architectural changes. For instance, in image–text retrieval, one could use CLIP to generate image tokens and reduce their number via average pooling or other compression methods. A MoME layer could then be inserted in parallel with CLIP's transformer layers, enabling elastic retrieval based on resource constraints. Training would proceed by computing the task-specific loss (e.g., contrastive loss) across different token compression rates, similar to how we use multiple audio-visual granularities in AVSR.

## G Limitations

While MoE-based models limit inference-time computation by activating only a small subset of experts, they still require all experts to reside in memory, resulting in increased memory usage. Since MoME follows the MoE paradigm, it inherits this memory overhead. However, because our experts are lightweight due to the bottleneck design, the overall memory footprint remains modest. Additionally, the inference cost of MoME is slightly higher than baselines like Llama-AVSR and Llama-MSTK. Finally, our paper focuses exclusively on audio-visual models. However, due to

its flexible design, MoME can be readily integrated into other LLMs and applied to tasks beyond audio-visual learning, such as vision-language modeling. Exploring its effectiveness in these broader domains is a promising direction for future work.

## H  Societal Impacts

MoME utilizes pre-trained large language models (LLMs), which inherently carry the limitations of LLMs, such as the potential to generate inaccurate or biased outputs. Therefore, we advise caution and recommend conducting thorough safety and fairness evaluations before deploying MoME in any downstream applications.

