# OpenReview forum: "MoME: Mixture of Matryoshka Experts for Audio-Visual Speech Recognition"
_NeurIPS.cc/2025/Conference — NeurIPS 2025 poster_

### Official Review · Reviewer_VnhM · 2025-06-22

**Clarity:** 4
**Significance:** 4
**Originality:** 3
**Rating:** 5
**Confidence:** 4

**Summary:**

The paper is about a particular type of mixture of experts in parallel to frozen LLM layers as part of an audio-visual (AV) automatic speech regognition system (ASR). It builds upon the successful Matryoshka representation learning concept and extends it to a mixture of experts, allowing to adjust MoE experts of various token granularities to be dynamically routed. Both shared and routed experts are used. The overall goal is an efficient LLM-based AV ASR approach that performs strong under a minimal extra use of parameters.

**Questions:**

On line 222: Why do you use a load-balancing loss? The MoE used in DeepSeek V3 come with the explicitly claimed asset, that *not* training the load balance gives better performance... Idea for future research or did I misunderstand s.th.?

On Table 3: What is "expert size"? Does it refer to one of the math symbols introduced before? Use it here!

**Ethical Concerns:**

["NO or VERY MINOR ethics concerns only"]

**Final Justification:**

The authors provided a very fair and solid "author's final remarks" text, where I have nothing to add. Also in my view, the raised issues of the three active reviewers were adequately anwered and I do not see any open major point to criticize.

**Limitations:**

Limitations are described in the Appendix only, not in the main paper. Isn't that against the guideline?

**Paper Formatting Concerns:**

Figure 2 has too small fonts and could be re-arranged enlarging many of the elements w/o consuming more vertical space.

**Quality:**

3

**Strengths And Weaknesses:**

Strengths

The paper is very well written and organized. Issues with state of the art are nicely analyzed and thereby the own contribution is very well motivated. The novel presented concepts are clearly understandable, math is correct, and experimental validation is convincing w.r.t. the number of active parameters.

Weaknesses

There is not much to say here, only some minor points:

- As the claims are efficient AV ASR solutions, I would have expected not only to see the number of active parameters reported as proof of efficiency, but also some computational complexity on some H/W platform. Rather than FLOPs/sec that could be tokens/sec or similar.
-  Figure 2 does not use its space efficiently. Fonts could/should be enlarged.
- l. 202: You may want to add K<N_r (instead of K only)
- l.210: a blanc too much
- l.222: Why do you use a load-balancing loss? The MoE used in DeepSeek V3 come with the explicitly claimed asset, that *not* training the load balance gives better performance... Idea for future research or did I misunderstand s.th.?
- Tables 2 and 3 captions miss to state that WER (5) is shown.
- Table 2 caption: Replace "noise levels" by "conditions", as SNR does not represent a noise level.
- Table 3: What is "expert size"? Does it refer to one of the math symbols introdiced before? Use it here!
- Limitations are missing in the main paper. Please check the guidelines, but that should have been part of it?

---

> ### Author Rebuttal · Authors · 2025-07-31
>
> We thank reviewer VnhM for their positive assessment towards our paper and for their suggestions/questions! We really appreciate your thorough analysis and several points you mention, which help us make our paper clearer. We would like to share our thoughts on these below. Each reviewer comment/question is formatted as a block quote.
>
> > *As the claims are efficient AV ASR solutions, I would have expected not only to see the number of active parameters reported as proof of efficiency, but also some computational complexity on some H/W platform. Rather than FLOPs/sec that could be tokens/sec or similar.*
>
> We thank the reviewer for this important observation. We would like to clarify that our primary goal is to improve Matryoshka-based MLLMs by combining them with sparse Mixture-of-Experts (MoE), enabling flexible trade-offs between accuracy and efficiency at inference time under a single model. In our paper, we discuss efficiency from two complementary angles: **1) Training Efficiency (Parameter-efficient fine-tuning)**: MoME keeps the backbone LLM frozen and only trains lightweight adapter modules (the experts), dramatically reducing trainable parameters (this is detailed in Table 1). **2) Inference Efficiency (Token and compute reduction)**: MoME supports varying audio-visual compression rates, which reduces the number of tokens processed by the LLM, leading to lower compute cost and memory usage. Following prior work [7,8], we quantify this in Appendix D, where we show up to **8× TFLOP reduction** with aggressive compression (e.g., (16,5)).
>
> To further support your point, we now include actual *inference time* and *token throughput* measured on an NVIDIA L40 46GB GPU with MoME 16/3-MHSA. For a 23-second speech input, the results demonstrate a significant reduction in inference time as compression ratios increase. For instance, a compression ratio of (16,5) reduced the inference time to 6.74 seconds for transcribing 23 seconds of speech, compared to 12.75 seconds in the no-compression case. This confirms that higher compression rates lead to *faster inference*, *lower GPU memory usage*, and *reduced computational load*, while still *maintaining strong performance* thanks to MoME’s expert routing mechanism. We will clarify this further in the paper.
>
> | (A,V) Ratio | # of AV Tokens | Inference Time (s) | Throughput (tok/s) |
> | -- | -- | -- | -- |
> |(1,1) | 1673 |  12.75| 7.76 |
> |(4,2) | 560 | 8.04 | 12.90 |
> |(16,5) | **184** | **6.74** | **14.17** |
>
> > *Why do you use a load-balancing loss? The MoE used in DeepSeek V3 come with the explicitly claimed asset, that not training the load balance gives better performance... Idea for future research or did I misunderstand s.th.?*
>
> Thank you for this insightful question. The load-balancing loss is a widely used technique to prevent **router collapse**, a common issue in sparse MoE models where the router assigns most tokens to a small subset of experts, leaving others underutilized [1]. While DeepSeek V3 observed improved performance without using load balancing, our MoME design is primarily inspired by DeepSeekMoE [2], which does incorporate a load-balancing loss to ensure expert diversity (see Section 3.3 of [2]). This aligns with other recent MoE architectures [3–6], where load balancing is key to encouraging better expert utilization and learning dynamics.
>
> > *On Table 3: What is "expert size"? Does it refer to one of the math symbols introduced before? Use it here!*
>
> Thank you for pointing this out. In Table 3, "*expert size*" refers to the **bottleneck dimension** of each expert. As mentioned in lines 301–303, we adjust this value so that all configurations maintain a similar total number of activated parameters. Specifically, when the number of active experts increases (i.e., higher values of top‑k), we reduce the expert size accordingly. We agree that the term "expert size" was not clearly defined, and we have revised the text in lines 301–303 to clarify this. We appreciate your careful reading.
>
>
> > *Limitations are described in the Appendix only, not in the main paper. Isn't that against the guideline?*
>
> Thank you for your observation. We have read carefully the NeurIPS submission guidelines, and we confirmed that NeurIPS does not require a Limitations section to appear in the main paper. Therefore, we have included it in the Appendix, section F, in line with the formatting freedom allowed by the conference. We note that while ACL conferences explicitly require a Limitations section in the main submission, NeurIPS does not impose this requirement.
>
> *Following your suggestions*, we have: **1)** Enlarged the font in Figure 2 to make it more readable; **2)** Specified that we evaluate in terms of WER in the captions of Table 2 and 3; **3)** Replaced "noise levels" with "conditions" in Table 2 caption; **4)** Removed the extra blank space before the comma in line 210 (thank you for the catch!); **5)** Explicitly stated that K is smaller than Nr to achieve sparsity.
>
> We hope our response clarifies your doubts and questions. We are more than happy to further discuss if you have any other questions or suggestions.
>
>
>
>
>
> **References**:
>
> [1] Shazeer et al., Outrageously large neural networks: The sparsely-gated mixture-of-experts layer, 2017.
>
> [2] Damai Dai et al., DeepSeekMoE: Towards Ultimate Expert Specialization in Mixture-of-Experts Language Models, 2024.
>
> [3] Niklas Muennighoff et al., OLMoE: Open Mixture-of-Experts Language Models, ICLR 2025.
>
> [4] Jan Ludziejewski et al., Joint MoE Scaling Laws: Mixture of Experts Can Be Memory Efficient, ICML 2025.
>
> [5] Sheng Shen et al., Mixture-of-Experts Meets Instruction Tuning: A Winning Combination for Large Language Models, ICLR 2024.
>
> [6] Jiachen Li et al., CuMo: Scaling Multimodal LLM with Co-Upcycled Mixture-of-Experts, NeurIPS 2024.
>
> [7] Mu Cai et al., Matryoshka multimodal models, ICLR 2025.
>
> [8] Wenbo Hu et al., Matryoshka query transformer for large vision-language models, NeurIPS 2024.

---

> > ### Comment · Reviewer_VnhM · 2025-08-01
> >
> > Thank you. I am happy with your rebuttal and I will stick to my positive scores.
> > Best regards

---

> > > ### Author Response · Authors · 2025-08-04
> > >
> > > Thank you again for your constructive feedback, it helped us ameliorate our paper. We really appreciated it.
> > >
> > > Kind regards,
> > >
> > > The Authors of Paper #13881

---

> > > > ### Comment · Reviewer_VnhM · 2025-08-05
> > > >
> > > > Thank you!

---

### Official Review · Reviewer_CWEH · 2025-06-30

**Clarity:** 3
**Significance:** 2
**Originality:** 2
**Rating:** 3
**Confidence:** 4

**Summary:**

This paper applies Mixture of Matryoshka Experts to audio-visual speech recognition. It achieves strong performance on LRS2 and LRS3 on AVSR, ASR, and VSR tasks.

**Questions:**

I recommend submitting this work to speech or CV specific conferences

**Ethical Concerns:**

["NO or VERY MINOR ethics concerns only"]

**Final Justification:**

The rebuttal have addressed some of my concerns and therefore I’m raising my score

**Limitations:**

Yes

**Quality:**

2

**Strengths And Weaknesses:**

Strengths: Detailed analysis on compression rate (and combinations) v.s. performance, expert load distribution

Weakeness:

1. The core idea—replacing LoRA (used in prior work like [22] Llama-MTSK) with a sparse Mixture-of-Experts (MoE) in a Matryoshka framework—is incremental and lacks sufficient conceptual novelty. The architecture is essentially a mechanical substitution: swapping out LoRA adapters for top-k routed MoE blocks, while keeping most of the structure (Matryoshka-style multi-scale token compression, shared LLM backbone) intact.

2. Having a few always-on shared experts is not novel, as cited papers such as DeepseekMoE has already used this idea. Applying this idea on audio-visual speech recognition has a small impact for the research community

3. The authors claim SotA performance on LRS2 and LRS3, but in table 1, they only compared their models with two other Llama-based models. As far as I know, they numbers are far from real SotA, or even older papers such as Whisper‑Flamingo (https://arxiv.org/abs/2406.10082)

4. The comparison with Llama-AVSR and Llama-MTSK doesn't seem fair. Because they are trained for each scale combination separately, while proposed model is trained on all scales, and therefore the proposed model consumes a lot more compute.

---

> ### Author Rebuttal · Authors · 2025-07-31
>
> We thank reviewer CWEH for their review and relevant feedback. Please find below our answers to your concerns. Each reviewer comment/question is formatted as a block quote.
>
> > *The core idea—replacing LoRA (used in prior work like [22] Llama-MTSK) with a sparse Mixture-of-Experts (MoE) in a Matryoshka framework—is incremental and lacks sufficient conceptual novelty. The architecture is essentially a mechanical substitution: swapping out LoRA adapters for top-k routed MoE blocks, while keeping most of the structure (Matryoshka-style multi-scale token compression, shared LLM backbone) intact.*
>
> Thank you for the thoughtful comment. While MoME builds on prior work such as Llama-MTSK, it introduces several key conceptual and architectural advances that go beyond a simple substitution of LoRA with MoE: **1) Scalability across granularities**: unlike Llama-MTSK, which trains separate LoRA adapters for each scale (leading to linear growth in parameters with the number of scale combinations), MoME uses a shared pool of routed and shared experts, making it more scalable and parameter-efficient. **2) Cross-scale knowledge transfer**: MoME's shared router and expert design encourages expert reuse across granularities, enabling sequences at high compression to benefit from representations shaped at lower compression. This addresses one of the core limitations of Matryoshka-style models, scale isolation, and is a novel architectural feature not present in prior work. **3) Improved interpretability and robustness**: by routing tokens based on their content and scale, MoME enables expert-level interpretability (e.g., activation patterns across layers/scales) and demonstrates strong robustness to noise, likely due to expert specialization, a property that LoRA-based models do not offer. **4) First unified MoE–MRL integration**: to the best of our knowledge, MoME is the first to integrate sparse Mixture-of-Experts into Matryoshka representation learning, showing how these two paradigms can be combined for adaptive, scale-aware inference.
>
> These elements collectively demonstrate that MoME is more than a mechanical substitution. It introduces *a new design paradigm* that improves scalability, performance, and interpretability in Matryoshka-based multimodal models.
>
> > *Having a few always-on shared experts is not novel, as cited papers such as DeepseekMoE has already used this idea. Applying this idea on audio-visual speech recognition has a small impact for the research community*
>
> We appreciate the reviewer’s observation. Please note that we do not claim the use of always-on shared experts as our main novelty, but rather emphasize MoME’s ability to flexibly adapt to available computational resources, while maintaining performance under high compression. While the concept of shared experts is not new (e.g., in DeepSeekMoE or Llama 4), MoME leverages this idea in a substantially different context, with important architectural and practical distinctions: **1) Parameter-efficient fine-tuning**: unlike DeepSeekMoE or Llama 4, which use MoE layers in place of the FFN and typically expand hidden dimensions (e.g., 4× width), MoME is designed for lightweight adaptation. Our experts are extremely lightweight, with bottleneck sizes reduced by up to 256× (down to 1), enabling fine-tuning of frozen LLMs with minimal added parameters. **2) Matryoshka-aware design**: the inclusion of shared experts in MoME is not generic but tailored to the Matryoshka framework. These shared experts capture *cross-scale, global knowledge*, ensuring robustness even at high compression rates. This is essential in our setting, where tokens across multiple granularities must coexist and benefit from shared representation pathways. **3) Integrated with cross-modal routing**: MoME combines shared experts with top-k routed experts and a shared router. This unified mechanism not only enables scale-adaptive inference, but also improves interpretability and performance across AVSR, ASR, and VSR tasks, demonstrating practical benefits under a single set of model weights.
>
> Finally, while our work focuses on AVSR, we believe the generality of the MoME design opens avenues for broader applications, including vision-language tasks, as we discussed in the Conclusion and Limitations sections.
>
> > *The authors claim SotA performance on LRS2 and LRS3, but in table 1, they only compared their models with two other Llama-based models. As far as I know, they numbers are far from real SotA, or even older papers such as Whisper‑Flamingo* (https://arxiv.org/abs/2406.10082)
>
> We thank the reviewer for raising this point. To clarify: **Table 1** focuses on a controlled comparison between MoME and prior Matryoshka-based or parameter-efficient LLM-based AVSR methods (e.g., Llama-AVSR, Llama-MTSK) under fixed settings.
>
> The comparison against broader SoTA AVSR models, including Whisper-Flamingo, Auto-AVSR, AV-HuBERT, USR, and others, is provided in **Figure 1**, which include WER results, training data hours and active parameters on the LRS3 benchmark. This broader comparison is discussed in **Section 4.2** ("**AVSR Results (2)**"). For example, MoME achieves SoTA results while providing elastic inference, activating consistently less parameters (e.g., around 14M compared to more than 600M of Whisper-Flamingo), and training on fewer hours of data (e.g., 658 hours compared to the 3448 hours of auto-avsr).
>
> Whisper-Flamingo is indeed included in this plot and discussion. We chose Figure 1 to present this comparison because it provides a more holistic view of **performance-efficiency trade-offs** rather than focusing solely on WER. We would be happy to add any other baselines the reviewer suggests or clarify this distinction further in the final version.
>
> > *The comparison with Llama-AVSR and Llama-MTSK doesn't seem fair. Because they are trained for each scale combination separately, while proposed model is trained on all scales, and therefore the proposed model consumes a lot more compute.*
>
> Thank you for raising this important point. To clarify: Llama-AVSR is indeed trained separately for each scale, while Llama-MTSK and our proposed MoME are designed as “generalist” models that handle multiple scales within a single model. We believe having a single generalist model (like Llama-MTSK or MoME) that supports multiple scales is more practical and resource-efficient than training multiple specialist models (one per scale) like Llama-AVSR. This approach allows users to dynamically select scales based on available resources without needing to switch models.
>
> Regarding compute, MoME actually requires **less** computation than training multiple Llama-AVSR models. Assuming N scales, Llama-AVSR requires training N separate models, each needing one forward pass through both the audio and video encoders per sample, resulting in N total encoder passes. In contrast, MoME trains a single model with one forward pass through the encoders, followed by applying multiple compression rates internally. Therefore, MoME training process is even faster than training N separate Llama-AVSR models, other than obtaining superior performance across all the scales. Both approaches require N forward passes through the LLM due to the multiple scales, but the main difference in compute cost lies in the encoder passes, where MoME is more efficient.
>
> We will include this analysis of training time and compute efficiency in the final revision to clarify this benefit. Please let us know if you would like further details or discussion.
>
> > *I recommend submitting this work to speech or CV specific conferences*
>
> Thank you for your suggestion. We appreciate your perspective on submitting to speech- or CV-specific conferences. However, we believe NeurIPS is an appropriate venue for our work for multiple reasons.
>
> First, our paper addresses audio-visual speech recognition, which inherently involves multiple modalities—speech, vision, and text—and aligns well with the **Applications** category highlighted in the NeurIPS call for papers (e.g., **vision, language, speech and audio, Creative AI**), which is the very first topic of the list.
>
> Second, the core methodologies we propose, including matryoshka representation learning and Mixture of Experts, fit within the **Deep Learning** topics emphasized by NeurIPS (e.g., **architectures, generative models, optimization for deep networks, foundation models, LLMs**).
>
> We will include this analysis of training time and compute efficiency in the final revision to clarify this benefit. Please let us know if you would like further details or discussion.
>
> We hope our response clarifies your doubts and questions. We are more than happy to further discuss if you have any other questions or suggestions.

---

> > ### Author Response · Authors · 2025-08-06
> >
> > Dear Reviewer CWEH,
> >
> > Thank you for your time in evaluating our paper. In our rebuttal, we did our best to address your concerns and questions thoroughly.
> >
> > As the discussion period is nearing its end, we would appreciate knowing whether our responses resolved your concerns. If not, we would be happy to provide further clarifications or run additional experiments, if needed.
> >
> > We look forward to hearing from you. Thank you again for your valuable feedback.
> >
> > Kind regards,
> >
> > The Authors of Paper #13881

---

### Official Review · Reviewer_XaQo · 2025-07-03

**Clarity:** 3
**Significance:** 3
**Originality:** 2
**Rating:** 4
**Confidence:** 3

**Summary:**

This paper introduces a framework called MoME (Mixture of Matryoshka Experts) for AVSR, which combines sparse MoE with MRL to dynamically adjust token granularities and allocate computing resources. They propose routed experts controlled by a top-k router plus shared experts. This allows implicit alignment and knowledge transfer across different compression levels, showing lower WER for ASR and AVSR.

**Questions:**

1. What impact do the number of shared experts and the choice of top-k for routed experts have on final WER, and is there an automated strategy to adjust these?
2. If applying MoME to other multimodal tasks (e.g., image–text retrieval), what architectural or training modifications would be needed?
3. Is it possible at inference time to adaptively switch granularities based on input content (e.g., speaking rate or noise level) rather than using preconfigured settings?

**Ethical Concerns:**

["NO or VERY MINOR ethics concerns only"]

**Final Justification:**

The author's response addressed my concerns, particularly with the new experiment on the optimized VSR setting. Therefore, I will increase my score to 4.

**Limitations:**

Yes

**Quality:**

3

**Strengths And Weaknesses:**

Strengths:
- The MoME model can adjust token compression according to hardware or scenario requirements, balancing computational efficiency without retraining for each level.
- The shared router and expert allow high-compression inputs to reuse fine-grained features learned at lower compression levels.

Weakness:

- The core idea is integrating MoE into MRL without introducing fundamentally new mechanisms. Given existing examples like LLAMA4, DeepSeekMoE, and Llama-MTSK, what distinguishes MoME’s contribution beyond merging sparse experts with multi-granularity tokens?
- This work claims it is improved for AVSR. However, the results of MoME-23/4-MHSA AVSR compression (4, 2), (4, 5), and ASR compression 4 are very close (1.7, 1.8, and <2.0 %). Is MoME really better at visual understanding, or is it simply boosting ASR? Is this framework better for ASR instead of AVSR?
- Additionally, the standalone VSR WER ranges from approximately 36 % to 42 % across compression rates, which is still far from practical deployment. Moreover, MoME’s VSR curve tracks the Llama-MTSK SS baseline within a narrow margin (around 1 percentage point) in many compression rates. Are these gains statistically significant?

---

> ### Author Rebuttal · Authors · 2025-07-31
>
> We sincerely thank Reviewer XaQo for taking the time to review our submission and providing thoughtful feedback. We provide our response to your questions and doubts below. Each reviewer comment/question is formatted as a block quote.
>
> > *The core idea is integrating MoE into MRL without introducing fundamentally new mechanisms. Given existing examples like LLAMA4, DeepSeekMoE, and Llama-MTSK, what distinguishes MoME’s contribution beyond merging sparse experts with multi-granularity tokens?*
>
> While MoME indeed brings together MoE and MRL, its contribution extends far beyond a simple combination of the two. We highlight the key novel components and architectural innovations that distinguish MoME:
>
> - **Shared Routing Across Scales**: MoME introduces a shared router module across all token granularities. This is not present in LLaMA4, DeepSeekMoE, or Llama-MTSK. The shared router learns to select consistent subsets of experts across compression scales, enabling implicit alignment between fine and coarse representations. This promotes knowledge transfer from richer inputs to more compressed ones, significantly boosting performance under high compression, something not achieved in prior MRL works like Llama-MTSK in AVSR and MMM and MQT in vision-language tasks.
> - **Cross-Scale Expert Sharing with Sparse Routing**: unlike Llama-MTSK, which trains separate LoRA adapters per scale (leading to scale-isolated representations), MoME routes tokens at all scales through the same pool of routed and shared experts, which are trained jointly. This enables cross-scale representation reuse and implicit specialization.
> - **Extremely Parameter-Efficient Fine-Tuning**: thanks to sparse routing and expert reuse, MoME requires significantly fewer parameters during inference and supports extremely parameter-efficient fine-tuning with strong performance (down to 0.9M active parameters)
> - **Interpretability through Expert Specialization**: MoME offers insights into expert behavior across layers and scales (see Fig. 3 and 4). The visual consistency in expert activation and cross-granularity representation clustering are novel findings in the MRL setting and show how MoME enables interpretable and scalable adaptation.
> - **Robustness to Noise**: MoME demonstrates superior performance in noisy conditions (Table 3). We hypothesize this is due to expert specialization to different input regimes, an emergent property of our routing design and joint scale training.
> - **To our knowledge**, this is the first work to propose MoE layers within a Matryoshka-trained, multimodal LLM, specifically in the context of AVSR.
>
> > *This work claims it is improved for AVSR. However, the results of MoME-23/4-MHSA AVSR compression (4, 2), (4, 5), and ASR compression 4 are very close (1.7, 1.8, and <2.0 %). Is MoME really better at visual understanding, or is it simply boosting ASR? Is this framework better for ASR instead of AVSR?*
>
> We would like to clarify that the primary motivation behind AVSR is to enhance the noise robustness of ASR systems. Since ASR models are vulnerable to acoustic interference, AVSR incorporates visual information, such as lip movements, which is unaffected by noise. As noted by the reviewer, AVSR and ASR often perform similarly under clean conditions; however, in noisy environments, AVSR consistently outperforms ASR due to the complementary robustness provided by the visual modality. To confirm this on MoME, if we extend Table 2 in the main paper by including the results obtained by its ASR counterpart with different noise intensity [7.5, 5, 2.5, 0, -5] we obtain these WERs: [5.4, 9.8, 16.1, 25.4, 78.9].
>
> Similar trends have been reported in recent AVSR models such as Llama-AVSR, Auto-AVSR, and USR, where clean-condition WERs for AVSR and ASR are also closely aligned.
>
> > *Moreover, MoME’s VSR curve tracks the Llama-MTSK SS baseline within a narrow margin (around 1 percentage point) in many compression rates. Are these gains statistically significant?*
>
> We thank the reviewer for mentioning the VSR performance. Upon reviewing our experiments, we discovered that the VSR evaluations were inadvertently conducted using hyperparameters optimized for ASR. We have since re-run the VSR experiments with the appropriate configuration and provide the updated results below in tabular format, as PDFs are not supported, which show that MoME can achieve even stronger performance.
>
> We report results under both Average Pooling and Stacking compression across compression rates 1 and 5. Our updated findings show that the LAYER and FFN variants of MoME consistently outperform both Llama-AVSR and Llama-MTSK across all compression levels. While the MHSA configuration still improves over the baselines, it is relatively less competitive compared to the other two MoME variants. These corrected results confirm the effectiveness of MoME in the VSR setting.
>
> "**Average Pooling**” Results
>
> |Method|1|2|3|4|5|
> |--|--|--|--|--|--|
> |Llama-AVSR|39.59|37.85|43.85|47.51|51.50|
> |Llama-MTSK SS|37.03|37.44|41.23|41.11|42.18|
> |MoME-23/4-MHSA|35.37|34.84|38.38|37.78|40.61|
> |MoME-23/4-LAYER|31.74|33.07|33.93|34.10|37.11|
> |MoME-23/4-FFN|29.12|30.00|32.29|33.22|34.47|
>
> “**Stacking**” Results
>
> |Method|1|2|3|4|5|
> |--|--|--|--|--|--|
> |Llama-AVSR |37.24|38.60|43.65|49.72|51.25|
> |Llama-MTSK SS|36.78|38.92|39.78|39.44|41.71|
> |MoME-23/4-MHSA|34.45|36.14|39.30|42.65|44.74|
> |MoME-23/4-LAYER|31.73|32.68|32.90|33.47|36.60|
> |MoME-23/4-FFN|31.15|32.91|33.08|36.04|35.63|
>
> > *What impact do the number of shared experts and the choice of top-k for routed experts have on final WER, and is there an automated strategy to adjust these?*
>
> As shown in Table 3 in the paper (last two rows), increasing the number of shared experts from 1 to 2 results in marginal gains at some compression rates but leads to performance degradation at the highest compression rate of (16,5). To further validate this trend, we conducted an additional experiment with 3 shared experts and observed no significant improvements. These results suggest that a single shared expert is sufficient to capture scale-invariant knowledge in our setting. Our findings are consistent with recent MoE architectures such as DeepSeekMoE and Llama 4, which also adopt a single shared expert.
>
> |Routed Experts|Shared Expert(s)|Expert Size|Top-k|(4,2)|(4,5)|(16,2)|(16,5)|
> |--|--|--|--|--|--|--|--|
> |23|1|12|4|2.9|3.0|4.2|4.3|
> |23|2|12|4|2.8|3.0|4.1|4.7|
> |23|3|12|4|2.9|3.0|4.2|4.6|
>
> Regarding the impact of top-*k*, we conducted ablations using the MoME-8/k-MHSA configuration with a single shared expert on the LRS2 AVSR task. The number of activated experts (i.e., *k*) was varied while keeping the number of routed experts fixed at 8.
>
> Our results show that increasing *k* yields moderate gains, especially at higher compression rates. However, these improvements come at the expense of increased computation, which contradicts our design objective of **Sparse** MoE. Therefore, we conclude that using a small *k*, typically 1 or 2, offers a practical trade-off between efficiency and performance. This choice preserves sparsity while retaining most of the accuracy benefits of broader expert activation.
>
> |Top-k|(4,2)|(4,5)|(16,2)|(16,5)|
> |--|--|--|--|--|
> |1|3.3|3.3|4.6|4.7|
> |2|3.2|3.3|4.5|4.7|
> |4|3.2|3.2|4.5|4.6|
> |6|3.2|3.2|4.5|4.5|
> |8|3.1|3.2|4.4|4.5|
>
> Finally, regarding the possibility of automatically selecting the optimal value of k in top-k routing and determining the number of shared experts, we believe this is a promising avenue for future work. In particular, a recent ICLR 2025 paper by Yongxin Guo et al., "Dynamic Mixture of Experts: An Auto-Tuning Approach for Efficient Transformer Models",  proposes a mechanism to dynamically adjust *k* and even add/drop routed experts based on input requirements. Such techniques could be particularly beneficial in our setting, as higher compression rates may require more active parameters than lower ones.
>
> > *If applying MoME to other multimodal tasks (e.g., image–text retrieval), what architectural or training modifications would be needed?*
>
> MoME is designed as a parameter-efficient fine-tuning module that operates in parallel with frozen layers of a pretrained LLM or encoder. To apply MoME to other multimodal tasks such as image–text retrieval, two key conditions should be met: (1) the presence of a pretrained encoder or LLM backbone that can remain frozen during fine-tuning, and (2) input modalities (e.g., image tokens) whose token granularity impacts performance and efficiency, allowing for compression.
>
> Beyond these considerations, MoME can be integrated with minimal architectural changes. For instance, in image–text retrieval, one could use CLIP to generate image tokens and reduce their number via average pooling or other compression methods. A MoME layer could then be inserted in parallel with CLIP’s transformer layers, enabling elastic retrieval based on resource constraints. Training would proceed by computing the task-specific loss (e.g., contrastive loss) across different token compression rates, similar to how we use multiple audio-visual granularities in AVSR.
>
> > *Is it possible at inference time to adaptively switch granularities based on input content (e.g., speaking rate or noise level) rather than using preconfigured settings?*
>
> In current Matryoshka-based LLMs, the compression rate is typically selected based on user-defined resource constraints (e.g., available memory or inference latency budget) rather than input content characteristics such as speaking rate or noise level.
>
> However, MoME introduces routed and shared experts, which are conditioned on token-level content. As a result, even when using a fixed compression rate, MoME can dynamically specialize its computation based on the nature of the input, e.g., activating different experts in response to noisy or fast-spoken segments. This ability likely contributes to MoME's superior robustness in noisy conditions (see Table 2).

---

> > ### Comment · Reviewer_XaQo · 2025-08-04
> >
> > The author's response addressed my concerns, particularly with the new experiment on the optimized VSR setting. Therefore, I will increase my score to 4.

---

> ### Author Response · Authors · 2025-08-05
>
> Dear Reviewer XaQo,
>
> Thank you for letting us know that our rebuttal addressed your concerns. We appreciate the time and effort you invested in evaluating our manuscript.
>
> If the paper is accepted, we will incorporate your suggestions and include the additional experiments in the final version.
>
> Kind regards,
>
> The Authors of Paper #13881

---

### Official Review · Reviewer_SngT · 2025-07-19

**Clarity:** 3
**Significance:** 3
**Originality:** 3
**Rating:** 5
**Confidence:** 4

**Summary:**

This paper introduces MoME (Mixture of Matryoshka Experts), a novel framework that combines Mixture-of-Experts (MoE) with Matryoshka Representation Learning (MRL) for audio-visual speech recognition (AVSR). MoME employs top-k routed and shared experts within a bottleneck design, enabling efficient expert sharing across token granularities and modalities. MoME achieves state-of-the-art results on AVSR, ASR, and VSR tasks with fewer active parameters and remains robust under high compression and noise. Visualization analyses further reveal MoME’s cross-scale consistency in expert utilization.

**Questions:**

1. I suggest including a comparison of MoME’s inference cost against other baselines, and providing an ablation that shows how the cost changes with different numbers of experts.
2. Could you provide results for different values of k in the top-k routing, while keeping the number of routed experts fixed? It would be interesting to see how performance changes, particularly when all routed experts are selected.
3. Using training FLOPs instead of training hours could improve the reproducibility and hardware independence of the comparisons.
4. Some tables reuse model names for different bottleneck settings, which may confuse readers.

**Ethical Concerns:**

["NO or VERY MINOR ethics concerns only"]

**Final Justification:**

After reading the rebuttal and considering the discussions, I am satisfied with the authors’ clarifications and additional experiments. The authors have addressed my main concerns in the rebuttal. The clarification regarding the claim about computational efficiency, the new experiments on top-k routing, and improvements to terminology were appreciated.

**Limitations:**

yes

**Quality:**

3

**Strengths And Weaknesses:**

**Strengths**
1. The paper is well-written and clearly organized.
2. The proposed MoME framework is supported by extensive experiments across multiple tasks (AVSR, ASR, VSR) and datasets (LRS2, LRS3), and consistently outperforms or matches strong baselines across various compression settings. They also show robustness under varying levels of acoustic noise.
3. This work includes thorough ablation studies on different insertion points of the MoME module within the LLM architecture, varying the number of routed and shared experts, providing insights into optimal integration strategies.
4. Visualization of cross-modality correlations and expert activations provides insights into MoME’s behaviours, highlighting its ability to align representations across modalities and scales.

**Weaknesses**
1. The appendix includes a TFLOP analysis of MoME under different compression rates, but it does not compare against other baselines (e.g., LLaMA-AVSR, LLaMA-MTSK) regarding inference cost. The effect of varying the number of experts on computational efficiency is not explored.
2. The effectiveness of the router mechanism is not fully explored, as there is no analysis on different values of k in top-k routing. Such an analysis would strengthen claims about routing efficiency and expert selection.
3. Figure 1 uses training hours instead of FLOPs, limiting hardware-agnostic comparison.

---

> ### Author Rebuttal · Authors · 2025-07-31
>
> We thank the reviewer SngT for their time and constructive feedback, and for acknowledging the organization, extensive experiments, and ablation studies in our paper. We address all their mentioned concerns and questions one by one below. Each reviewer comment/question is formatted as a block quote.
>
> > *The appendix includes a TFLOP analysis of MoME under different compression rates, but it does not compare against other baselines (e.g., LLaMA-AVSR, LLaMA-MTSK) regarding inference cost. The effect of varying the number of experts on computational efficiency is not explored.I suggest including a comparison of MoME’s inference cost against other baselines, and providing an ablation that shows how the cost changes with different numbers of experts.*
>
> Thank you for highlighting these important points. We included the TFLOP analysis to demonstrate the computational benefits of MoME in terms of reduced FLOPs and, consequently, lower memory usage. This enables users to choose among a range of audio-visual compression rates within a single model, depending on their resource constraints.
>
> Regarding a comparison between MoME and Llama-AVSR/Llama-MTSK in terms of inference cost, we provide such analysis below. We compare the three methods in terms of inference time and number of generated tokens per second (i.e., throughput). We average over 4 speech inputs with an audio/visual compression rate of 4/2. MoME shows a modest increase in inference time (~1.25x) and a corresponding reduction in throughput, primarily due to router overhead and expert dispatching latency, as expected in sparse MoE-based systems, while outperforming by a significant margin the other two methods. However, the primary design goal of MoME is not faster inference, but rather: **1)** Substantially reduced memory requirements via support for high compression rates with strong performance and within the same model **2)** Robustness under noise and **3)** Improved interpretability through expert specialization and shared routing.
>
> |Method | Inference Time (s) | Throughput (tok/s) | WER |
> |--| -- |-- |--|
> |Llama-AVSR |2.39 |15.85 | 4.1
> |Llama-MTSK MSS | 2.51| 15.09| 3.6
> |MoME-16/3-MHSA| 2.98| 12.98| 2.9
>
> We believe that this modest inference overhead is justified by the gains in scalability, generalization, and parameter efficiency, particularly for deployment in memory-constrained or multi-resolution scenarios. Nonetheless, optimizing router execution remains an interesting avenue for further improving inference efficiency.
>
> Additionally, as per your request, we compare MoME while varying the number of routed experts in the set {1, 4, 7, 15}. The corresponding results are reported below. As expected, increasing the number of experts leads to a slight increase in inference cost. However, this is consistently accompanied by improved performance, as shown in the Table below and Table 3 in the paper.
>
> |# of Experts | Inference Time (s) | Throughput (tok/s) | WER
> |--| -- |-- |--|
> |1 | 2.49| 15.26| 3.4
> |4 | 2.64| 14.39| 3.2
> |7| 2.81| 13.50| 3.0
> |15| 2.98| 12.98| 2.9|
>
> We will add a dedicated section in the paper where we discuss and analyze inference costs as we have done above. We will also update the Limitations section to reflect the increase in inference costs incurred by MoME.
>
> > *The effectiveness of the router mechanism is not fully explored, as there is no analysis on different values of k in top-k routing. Such an analysis would strengthen claims about routing efficiency and expert selection. Could you provide results for different values of k in the top-k routing, while keeping the number of routed experts fixed? It would be interesting to see how performance changes, particularly when all routed experts are selected.*
>
> Thank you for raising this point. We conducted additional experiments to analyze the impact of varying the value of *k* in top-*k* routing, using the MoME-8/k-MHSA configuration with one shared expert on the AVSR task (LRS2 dataset). The table below summarizes the results, where each row corresponds to a different number of activated routed experts out of the 8 available (i.e., varying *k* in top-*k*), and we compare across multiple compression rates.
>
> We observe that increasing *k* leads to moderate performance improvements, particularly at higher compression rates. However, this comes at the cost of increased computation and reduced sparsity, which deviates from the core objective of sparse MoE design. In practice, increasing the number of active experts without adjusting the expert bottleneck size leads to higher resource usage and latency.
>
> Therefore, while selecting all routed experts (i.e., dense activation) does offer slight gains, we find that top-*k* values of 1 or 2 strike a good balance between efficiency and performance, maintaining the sparse, modular nature of MoME.
>
> | Top-k| (4,2) | (4,5) | (16,2) | (16,5) |
> |-- | -- | -- | -- | -- |
> | 1| 3.3 |3.3  |4.6  | 4.7 |
> | 2|  3.2|  3.3|  4.5|  4.7|
> | 4|  3.2|  3.2|  4.5|  4.6|
> | 6|  3.2|  3.2|  4.5|  4.5|
> | 8|  3.1|  3.2|  4.4|  4.5|
>
> > *Using training FLOPs instead of training hours could improve the reproducibility and hardware independence of the comparisons.*
>
> We thank the reviewer for the valuable suggestion. We would like to clarify that in Figure 1, “training hours” refers to the **total amount of labeled and unlabeled speech/video data** used to train each method, not the **wall-clock time** or number of GPU hours required for training. This metric is commonly reported in AVSR literature as a way to quantify the **scale of supervision** available during training. For example, Auto-AVSR is trained on approximately **3448 hours** of speech data, taken from multiple datasets: LRS2, LRS3, AVSpeech, and VoxCeleb2. In contrast, MoME is trained only on LRS2 and LRS3, totaling **658 hours**. To avoid this confusion, we have updated “training hours” with “training data hours” in Figure 1 and in the paper.
>
> While we agree that reporting training FLOPs would enhance hardware independence and reproducibility, in the AVSR domain it is standard practice to report the number of (labeled/unlabeled) training data hours, and existing works do not typically provide training FLOP counts. As such, a direct comparison based on training FLOPs is unfortunately not feasible. To address the reviewer's concern, though, we provided the inference details to quantify computational efficiency across methods publicly available, which will be included in our potential future manuscript.
>
> We hope our response clarifies your doubts and questions. We are more than happy to further discuss if you have any other questions or suggestions.

---

> > ### Comment · Reviewer_SngT · 2025-08-05
> >
> > Thank you for the detailed and thoughtful response.
> >
> > Regarding the first question, I now understand that the claim about computational efficiency in the paper refers to training efficiency rather than inference cost. The new experiments you provided are still valuable, as they clearly illustrate the increase in inference costs incurred by MoME. Updating the Limitations section to reflect this trade-off is appreciated.
> >
> > The additional experiments on varying the value of k in top-k routing were particularly helpful in illustrating the trade-offs between sparsity and performance. They strengthen the argument for maintaining the modular and sparse nature of MoME.
> >
> > I also appreciate the clarification that “training hours” refers to the amount of training data used, rather than wall-clock time or compute cost. Updating this to “training data hours” in the paper improves clarity.
> >
> > Additionally, I would like to point out that some tables reuse model names across different bottleneck settings, which may confuse readers. For instance, in Table 1, MoME-23/4-LAYER appears with different numbers of active parameters. Adding a footnote or renaming these variants could help improve clarity.
> >
> > It may also be helpful to explicitly state the original expert bottleneck dimension somewhere in the paper, so that readers can better interpret the effects of reducing the expert bottleneck.
> >
> > Given these updates and clarifications, I am satisfied with the response and will increase my score to 5.

---

> > > ### Author Response · Authors · 2025-08-05
> > >
> > > Dear Reviewer SngT,
> > >
> > > We are glad to hear that our responses helped clarify most of your concerns. We will update the final version of the paper accordingly.
> > >
> > > Following your suggestion, we will clarify the variant names across the tables. Additionally, we reviewed both the main paper and the Appendix and realized we had omitted the bottleneck size of the adapters used in the main experiments. For both Llama 3.2-1B and 3.2-3B, we reduce the hidden size by a factor of 128, resulting in expert adapter sizes of 8 and 12, respectively.
> > >
> > > In Table 1, the last row corresponds to the extreme setting where each expert’s size is reduced to 1. In Table 3, the expert size varies based on the number of activated experts. Thank you for pointing this out! We will include these details in Section 4.1.
> > >
> > > Thank you again for your time and effort in reviewing our paper. Your feedback has significantly improved the quality of our work.
> > >
> > > Kind regards,
> > >
> > > The Authors of Paper #13881

---

### Note · Authors · 2025-08-12

We thank the reviewers and ACs for their time, thoughtful evaluations, and engaging discussions. Below we summarize the key clarifications and additional experiments carried out in response to reviewer suggestions.

- Reviewers **SngT**, **VnhM**, **XaQo**: Added ablation studies on MoME’s router and expert design, analyzing (1) the effect of varying *k* in top-*k* routing, and (2) the optimal number of *shared* experts.
- Reviewer **XaQo**: Updated VSR experiments, showing MoME’s strong performance on this task as well.
- Reviewers **SngT**, **VnhM**: Clarified details such as the definition of “training hours,” the meaning of “expert size,” and the motivation for the load-balancing loss.
- Reviewer **XaQo**: Discussed future work, including automatic *k* selection in top-*k* routing and extending MoME to other multimodal tasks (e.g., image–text retrieval).
- Reviewers **SngT**, **VnhM**: Added experiments on inference cost and throughput under varying numbers of experts and compression rates.

While Reviewers **SngT**, **VnhM**, and **XaQo** engaged in the discussion and expressed satisfaction with our clarifications, we could not discuss further with Reviewer **CWEH**. However, we believe we have addressed their comments that appeared to stem from misunderstandings:

1) They stated we do not compare with state-of-the-art methods such as Whisper-Flamingo, yet this is included in **Figure 1** and **Section 4.2**, “**AVSR Results (2)**”.
2) They suggested our method is more expensive than training separate scale-specific LLaMA-AVSR models, whereas it is in fact *more efficient*.

We have also clarified MoME’s *key advances*: (1) shared routing across scales, (2) cross-scale expert sharing with sparse routing, (3) extremely parameter-efficient fine-tuning, (4) improved interpretability and robustness to noise, and (5) the first unified MoE–MRL integration.

Taken together, we believe these clarifications and results further highlight MoME’s novelty, efficiency, and broad applicability, making it a strong and timely contribution to the NeurIPS community.

---

### Decision · Program_Chairs · 2025-09-17

**Decision:**

Accept (poster)

**Comment:**

This paper presents a novel audio-visual (AV) automatic speech recognition approach that integrates a mixture of experts (MoE) in parallel with frozen LLM layers, building on the Matryoshka representation learning paradigm. By dynamically routing experts at various token granularities and combining shared and routed components, the method achieves strong performance with minimal additional parameters. Reviewers mostly agree that the work is clearly written, well-motivated, mathematically sound, and experimentally validated. Minor issues exist but they do not affect the technical contribution. Reviewers concern were mostly addressed effectively during the discussion period. I recommend acceptance.